



# Assessing the capabilities of the SWOT mission for large lake water surface elevation monitoring under different wind conditions

Jean Bergeron[1], Gabriela Siles[1], Robert Leconte[1], Mélanie Trudel[1], Damien Desroches[2], Daniel L. Peters[3]

5 [1]Département de génie civil, Faculté de génie, Université de Sherbrooke, 2500 Boul. Université, Sherbrooke, QC J1K 2R1, Canada
[2]Centre National d'Études Spatiales (CNES), 31400 Toulouse, France
[3]Watershed Hydrology and Ecology Research Division, Environment and Climate Change Canada, University of Victoria, Victoria, BC, V8W 3R4, Canada

10 *Correspondence to*: Jean Bergeron (Jean.Bergeron2@USherbrooke.ca)

**Abstract.** Lakes are important sources of freshwater and provide essential ecosystem services. Monitoring their spatial and temporal variability, as well as of their functions, is an important task within the development of sustainable water management strategies. The Surface Water and Ocean Topography (SWOT) mission will provide continuous information on the dynamics of continental (rivers, lakes, wetlands and reservoirs) and ocean water bodies. This work aims to contribute to 15 the international effort evaluating the SWOT satellite (2022 launch) performance for water balance assessment over large lakes (e.g., >100 km$^2$). For this purpose, a hydrodynamic model was set up over Mamawi Lake, Canada, and different wind scenarios on lake hydrodynamics were simulated. The derived water surface elevations (WSE) were compared to synthetic elevations produced by the Jet Propulsion Laboratory (JPL) SWOT high resolution (SWOT-HR) simulator. Moreover, water storages and net flows were retrieved from different possible SWOT orbital configurations, as well as synthetic gauge 20 measurements. In general, a good agreement was found between the WSE simulated from the model and those mimicked by the SWOT-HR simulator. Depending on the wind scenario, errors ranged between approximately -2 and 5 cm for mean error, and 30 to 70 cm root mean square error. Low spatial coverage of the lake was found to generate important biases in the retrievals of water volume or net flow between two satellite passes in the presence of local heterogeneities in WSE. However, the precision of retrievals was found to increase as spatial coverage increases, becoming more reliable than the 25 retrievals from 3 synthetic gauges when spatial coverage approaches 100 %, demonstrating the capabilities of the future SWOT mission in monitoring dynamic WSE for large lakes across Canada.

## 1 Introduction

Inland freshwater systems (e.g., rivers, lakes, ponds, and wetlands) are important sources for society, as well as provide essential habitat to sustain biodiversity and valuable ecosystem services (e.g., Palmer et al., 2015). As part of the 30 hydrological cycle, the extent and volume of water stored on the landscape changes through time, which according to the water budget equation depends on inflow (precipitation, overland runoff, groundwater) and outflow (evaporation, seepage,



withdrawals to satisfy residential, agriculture and industrial demands, streamflow) components. Temporal monitoring of continental waters is important to assess trends and variability of their availability in a changing climate (e.g., Frasson et al., 2017; Bonsal et al., 2019), as well as to support programs for the mitigation of hydro-hazards (e.g., drought, flooding;

Rahman and Di, 2017). Hydrometric stations can provide useful information for water management endeavours (e.g., water levels, discharge). However, as stated in previous studies, the available network of gauges is spatially insufficient for the surveillance of global rivers and lakes (e.g., Pavelsky et al., 2014). Densifying this ground-based monitoring system would be costly and challenging to install and maintain in difficult-to-access areas due their remote location and/or in zones affected by political or other conflicts (wars, internationally shared waterbodies; e.g., Gleason and Hamdan, 2017).

Moreover, local measurements systems can present technical problems that translate into important gaps in the recorded time series. For example, extensive areas of the North America and Eurasia are affected by spring ice breakup events that can disrupt monitoring until the gauge is reset.

Remote sensing technologies and methods offer the possibility to complement and enhance in situ observations, providing valuable spatially distributed information on ungauged water systems (e.g., Santos da Silva et al., 2014). Delineation of and

changes in surface water area for river, wetland, and lake systems have been successfully achieved via the application of optical (e.g., Gardelle et al., 2010; Jiang et al., 2014), radar ( e.g., Ding et al., 2011; Zeng et al., 2017), and a combination of satellite imagery (e.g., Töyra et al., 2002, Bwangoy et al., 2010; Bioresita et al., 2019). The estimation of other important hydrological variables such as water level, discharge, water volume and their variations can represent a more difficult challenge (Grippa et al., 2019). This challenge has been addressed by exploiting information acquired by radar and laser

altimetry (e.g., Maheu et al., 2003; Crétaux and Birkett, 2006). For example, Santos Da Silva et al. (2012) used ENVISAT Radar Altimeter 2 (RA-2) data to characterize water storage in rivers, floodplains, wetlands and lakes in the Amazon basin. Internationally, estimations of changes in lake and reservoir volume have been attempted by exploiting different satellite altimetry databases including both laser (Ice, Cloud, and Land Elevation Satellite; ICEsat-1) and radar data (Topex/Poseidon, Jason-1, Jason-2, ENVISAT; e.g., Duan and Bastiaanssen, 2013; Crétaux et al., 2016). In Canada, Interferometric Synthetic

Aperture Radar (InSAR) has been employed to estimate relative water level changes for lakes and wetlands (e.g., Mohammadimanesh et al., 2018; Siles et al., 2020). Musa et al. (2015) present a review of hydrological applications using optical, radar imagery and altimetric data.

Despite the demonstrated potential of current radar altimetry satellite missions, the large footprint dimension (few hundred meters to several kilometers) makes difficult the detection of water levels for small water bodies (e.g., Grippa et al., 2019).

For example, in order to detect the water level over a given lake, the strong backscatter signal coming from several surrounding elements (nearby wetlands, small lakes, rivers, wet outcrops and sands) needs to be discriminated and removed from the measured waveform; otherwise it may not be possible to measure water levels from the target water body. IceSAT 1 and 2 laser sensors provide a better ground track spatial resolution allowing to partially overcome limitations of the radar sensors (e.g. Wang et al., 2011). However, the presence of clouds and other atmospheric effects reduce the accuracy of the

laser-based sensors (Brenner et al., 2007). In addition, the ground separation between successive tracks are, generally, very





large (several kilometers) and their revisit time can be infrequent (up to 35 days), which could limit the spatial and temporal information over a considered surface water feature. Some studies have successfully combined synthetic aperture radar (SAR) imagery and radar altimetry in order to over come these constraints and to provide high-resolution water levels (e.g., Baup et al., 2014) and water level changes (e.g., Kim et al., 2009). Despite the performance of those approaches, it remains
difficult to find altimetric data, radar and optical imagery acquired at the same time or within a few days difference for a given watercourse and body.

The Surface Water and Ocean Topography (SWOT) satellite mission, expected to be launched in 2022, is the first satellite of its type comprised of a bistatic near-nadir SAR interferometer that will enable to benefit from the combined advantages of radar altimeters (water level detection) and SAR imagery (high spatial resolution). This mission is a cooperation between the
NASA and the Centre National d'Études Spatiales (CNES), with collaboration of the Canadian Space Agency (CSA) and the United Kingdom Space Agency (e.g., Biancamaria et al., 2015). This Ka-band satellite will map surface water elevation (WSE) and water surface slope (WSS) along rivers and lakes around the globe, with SWOT derived sub products such as river discharge and lake water volume change (e.g., Biancamaria et al. 2015). Measurements from this satellite cover water surfaces (rivers wider than 100 m and lakes/reservoirs with a minimum surface area of 250 m x 250 m) located between 78°
S and 78° N with a revisit time of 21 days (Pavelsky et al., 2014). Due to the polar orbit of the satellite, lakes in higher latitude will likely benefit from multiple coverages per 21-day cycle.

Most published studies on SWOT capabilities have focused on its potential for river applications. For example, study the impact of different river-reach definition strategies on discharge estimation (Frasson et al., 2017), river depth assimilation (Häfliger et al., 2019) and river bathymetry definition (Yoon et al., 2012), among others applications (e.g., Garambois and
Monnier, 2015; Gleason et al., 2014; Domeneghetti et al., 2018, Oubanas et al., 2018). Only a few studies have been found in the literature that focused on the performance of SWOT over lakes and reservoirs. Munier et al. (2015) evaluated the effect of SWOT data assimilation on reservoirs, showing that it can contribute to improving the management of the Selingue Dam in the Niger River Basin, Africa, to meet environmental constraints. Solander et al. (2016) presented a characterization and estimation of errors that could affect SWOT observations on man-made reservoirs. Recently, Grippa et al. (2019)
showed the potential of SWOT for monitoring the seasonal variability of water levels and volume in small lakes of central Sahel, Africa.

Environmental factors, such as wind, can have a notable influence on the spatial distribution of water levels in large water bodies (e.g. lakes and oceans). For instance, high sustained winds from one direction can push the water to one end of the lake and drop at the opposite end. Such phenomenon, known as a seiche, resulted in surges reaching up to 3 meters in more
extreme cases in Lake Erie, eastern Canada (Farhadzadeh et al., 2017). In western Canada, wind seiches on Lake Athabasca have been observed, leading to vertical water fluctuations up to 1m and pushing water into adjacent connected lakes (e.g, Mamawi lake) of the Peace-Athabasca Delta (Timoney, 2013). The potential impacts of such wind generated episodic events on SWOT observations thus require investigation and understanding prior to mission launch.



To support the mission, the SWOT Canada Terrestrial Hydrology (SWOT-C TH) team has developed several research field-
and modelling-based projects (Pietroniro et al., 2019). One one of these projects is focused on the Peace-Athabasca Delta
(PAD) in northern Alberta, Canada. In support the international effort, the main goal of the following SWOT-C TH work is
to assess the capability of SWOT to quantify lake water volume changes for a large lake under different wind scenarios.
Specifically, the objective is to setup a hydrodynamic model over Mamawi Lake, located at the centre of the PAD in order
to: i) evaluate WSE as mimicked by the SWOT-High Resolution (HR) simulator to those provided by a hydrodynamic model
and in situ lake gauges; and ii) estimate water storages retrieved from different possible SWOT orbital pass coverage over
the lake.

## 2 Study area

The Peace Athabasca Delta (PAD; ~60000 km$^2$) is a RAMSAR Wetland site of international importance located in northern
Alberta, Canada, where the Peace and Athabasca River converge at the western end of Lake Athabasca (~7800 km$^2$). The
majority of the delta (80%) is located within the Wood Buffalo National Park, which is a UNESCO World Heritage Site. The
PAD encompass several large, relatively shallow, interconnected lakes (Lake Claire, Mamawi Lake, Richardson Lake) and
more than a thousand smaller lakes and wetlands that have varying hydraulic connectivity to the main flow system
depending on elevation and distance inland (Peters et al., 2006a; 2006b). The perched lakes and wetlands of the delta provide
important ecosystem services and are dependent on occasional floodwater input from ice-jam and open-water flood events to
maintain aquatic conditions. Climate variability and change, river regulation and water use in upstream basin areas influence
wetting and drying phases of the delta, which in turn influence local species, ecological functions and integrity (e.g., Beltaos,
2014; Ward and Gorelick, 2018; Bush et al., 2020).

At the heart of the PAD is Mamawi Lake (~200 km$^2$ and <3 m depth). This lake is directly connected to Lake Claire through
the Prairie River and Lake Athabasca through the Chenal des Quatre Fourches (Fig.1). A major source of water inflow is the
Athabasca River and outflow is normally northward toward the Slave River, but can reverse direction when stage on the
Peace River is higher than the central lakes, occasionally leading to extremely high lake levels that expand beyond the
shoreline (Leconte et al., 2001; Peters and Buttle, 2010). Mamawi lake is also prone to short term rise/fall in water levels due
to wind driven seiche events. Periodic seiches can move water into low-lying basins and expose mudflats, the dynamics of
which is poorly understood (Timoney, 2013).
The PAD receives continental arctic and maritime arctic air masses during the winter, whereas maritime polar winds are
common in the summer; northwesterly winds predominate most of the year (Phillips, 1990). The continental climate triggers
a wide variation of climatological variables between the winter and the summer (Phillips, 1990). Temperatures in the area
typically vary between < -20 °C and > 20 °C, with the highest temperature occurring during the month of July and the lowest
during the month of January (Peters et al., 2006a). The largest precipitations occur in the mid-summer and the lowest
amounts are usually at the end of winter (Peters et al., 2006a).



## 3 Data

A number of extensive ground-based and remotely sensed derived datasets are required to setup a hydrodynamic model and to perform the SWOT simulations. As part of the SWOT-C TH effort (Pietroniro et al., 2019), two fieldwork campaigns were carried out in the PAD during the summer 2017 by Environment and Climate Change Canada and the University of
Sherbrooke with support from Wood Buffalo National Parks. In addition to the long term water level hydrometric network stations on the Prairie River and Mamawi Lake outflow, project specific measurements of water level and streamflow were also taken in the main channels around Mamawi Lake (see Fig. 2). The bathymetry of the lake and channel cross-sections were acquired by a combination of weighted line depth and echo-sounding measurements. These data were used to define the boundary conditions and develop the hydrodynamic model (see Methodology section below).

A high-resolution Digital Elevation Model (DEM; 2 m) was developed using Light Detection and Ranging (LiDAR) elevation information provided by Environment and Climate Change Canada (ECCC)from the collection of aerial remote sensing campaigns in September 2012 and 2013, with missing areas covered by Shuttle Radar Topography Mission (SRTM). Details of DEM are provided in Siles et al. (2020). All elevation data were referenced to the NAD83 Canadian spatial reference system (CSRS) and Canadian Geodetic Vertical Datum (CGVD) 2013 epoch 2010. The DEM and bathymetry

were used to create the topobathymetry raster file incorporated in the hydrodynamic modelling.

The extent of Mamawi Lake is difficult to estimate precisely, both from remote sensing and in situ observations, due to the heavy vegetation and soft soil (mud) surrounding it, as well as the seasonal variability in the water level causing the lake to sometimes merge with neighbouring lakes to form a single water body in extreme cases. Nonetheless, a water mask that discriminated the water from the land pixels and used by the SWOT-HR simulator was determined from the combination of

a Sentinel-2 image acquired on July 9th 2017 and a Sentinel-1 image (Fig. 3a) acquired on July 8th 2017. This mask was also used to limit the physical boundaries of the lake in the hydrodynamic model. For the period considered in this study, the extent of the model is capped at 193 km² including reaches, while the inner lake extent is 175 km². The classification shown in Fig. 3b was estimated from the Sentinel-2 imagery using the Image Classification tool of the ArcGIS software. This classification was used to calibrate the Manning coefficients used in the hydrodynamic model (see Methodology section).

## 155    4 Methodology

### 4.1 Hydrodynamic simulations of Mamawi Lake

The 2D-H2D2 hydrodynamic model was used to analyze water motions beneath Mamawi Lake's surface. This finite element hydrodynamic software platform, developed at the "Institut National de la Recherche Scientifique, Centre Eau Terre Environnement" (INRS-ETE; (http://www.gre-ehn.ete.inrs.ca/H2D2; Secretan 2013), consists of multiple modules (e.g.,

sediment transport, water quality). H2D2 estimates the 2-D flow velocities, discharge, and water levels by solving the differential shallow water equations, also known as the Saint-Venant equations.



The triangular finite element mesh of the topobathymetry used in the hydrodynamic model was generated using the Surface Water Modelling System (SMS). SMS, developed by Aquaveo, was designed for 2D costal and riverine modelling. A total of 301,900 elements were used to model Mamawi Lake. Each of these elements has an elevation, which was obtained from the topobathymetric map generated from the field measurements, and a Manning roughness coefficient.

The Manning coefficients were estimated in a two-step process. First, a supervised classification was performed using Sentinel-2 data (Fig. 3b), which was validated using pictures taken from the ground, boat and helicopter. The classes include open water, shallow or flooded areas, algae, moss, sedge, live deciduous (mainly willows), denuded or dead willows (likely caused by successive tent caterpillar outbreaks; Gleeson, 2017) and evergreen vegetation. Since the classification matched relatively well our general knowledge of the lake configuration and the experiment is ultimately synthetic, it was not felt necessary to use a robust validation method traditionally used in remote sensing. The classification was instead used to infer starting Manning coefficient ratios for groups of classes. The classes open water, river, shallow and algae were grouped and given the same starting values of $n = 0.018$, a typical value for relatively smooth channels. The second step was to vary these coefficients in such a way as to match the water level conditions measured in the channels leading to Mamawi Lake assuming steady-state conditions. The coefficients for the other classes were also varied while keeping the same ratio as with the former group of classes. Coefficients for areas with the moss class were double compared with the first group, while the coefficients for remaining heavily vegetated classes (sedge, willows and evergreen) were four times those of the first group.

Once calibrated, the H2D2 simulations were assumed to represent the synthetic "true state" of Mamawi Lake. Several simulations were performed in transient mode over a period of 24 hours under various wind conditions. Every scenario began with the steady-state result under no wind (0 m/s) and ended after a day of constant wind speed and direction. The scenarios include wind speeds of 5, 10 and 15 m/s coming from different directions, including the cardinal and intercardinal points.

Homogenous wind speed and direction over the simulation period was a simplification of reality. However, true wind observations are not required as the results are not compared with real data. This also allowed for a simpler analysis of the simulation results and a first step toward a more thorough understanding of expected SWOT performance under more complex wind scenarios.

The H2D2 model used three boundary conditions located up (or down) the main channels leading to (or out) of the lake. Two of these were water levels, used at the Prairie River (west end) and Mamawi Channel (east end), while the third located closer to the lake at Mamawi Creek was set using a fixed inflow (Fig. 2). The boundary conditions were kept the same throughout the simulations. In reality, the wind could accelerate the flow downstream or slow the flow such that water accumulates upstream. However, the boundaries were specified a few kilometers outside the lake mask. This was done to reduce the impact of boundary conditions on lake water levels and avoid model divergence.



## 4.2 SWOT height simulations

From the four expected SWOT overpasses over Mamawi Lake, the one that covers the entire lake surface was selected (pass
#303; Fig. 1). The SWOT-HR simulator developed by NASA's JPL and installed at the CNES was used to simulate expected
height observations retrieved from SWOT. As input for the simulator the H2D2 simulated water heights and the mask
generated from the Sentinel-2 data, both integrated into a unique netcdf file, were used. Alternatively, the topobathymetry
and a water depth file can be used as inputs. Note that for our purposes, the high-resolution DEM was used as the truth and
the reference DEM, but if other types of analyses are targeted (e.g., impact of the topographic errors in the SWOT
observations) different quality DEMs could be used. From these inputs, interferograms (IFGs) were generated containing the
phase information from which the water levels were derived.

    The noise in the IFG that is later translated as the error height can be of two types. The first type of error is a gaussian noise
that is added to the simulated Single Look Complex (SLC) images and which comes from the instrument. This unbiased
instrumental noise corresponds to a mean average error of 4-5 cm per $km^2$ and can be reduced by an averaging procedure.
The second type of error is of deterministic nature and comes from the topographic layover. This error introduces a bias on
the estimated heights. As explained by Oubanas et al., (2018) this noise will depend on several factors such as the surface
type and the radar parameters. The noisy phases contained in the IFG are averaged by a procedure known as multilooking to
improve the Signal to Noise Ratio (SNR) at the expense of resolution (e.g. Ulaby et al., 2014). From the IFGs, pixels clouds
were derived by the simulator which were classified and geolocated. It is important to highlight that the output pixel cloud
can have important geolocation errors and that the synthetic noisy SWOT phases do not include other sources of perturbation
(e.g., wet and dry troposphere, ionospheric effects) that can disturb the actual signal of the satellite (Domeneghetti et al.,
2018), as reported for other altimeters (e.g., Frappart et al., 2015). Among the non-simulated errors, the residuals from the
roll error that are not corrected are the ones that can have a more critical impact. At the lake scale, this error can be
considered as a bias.

In order to perform the classification of the pixel cloud, the simulator uses preliminary defined classes based on the values of
the radar backscatter coefficient $\sigma_0$. Based on previous studies (e.g., Domeneghetti et al., 2018; Moller and Fernandez, 2015;
Fjørtoft et al. 2014), $\sigma_0$ values ranging from −5 to 10 dB and from 10dB to 15dB reasonably represent the signal coming from
land and water, respectively. The output pixel cloud is classified according to four principal classes (Land, Land near water,
Water near land, Interior water), with the possibility to add a fifth class corresponding to the 'dark water' (Peral et al., 2014).
The dark water pixels correspond to pixels that were not identified during the classification procedure because of a low local
backscattering signal (e.g., calm water). A priori Pekel mask (Pekel et al., 2016) is used in the processing to detect the
possible water areas and therefore supporting the identification of dark water patches. We emphasize that heights derived
from dark water pixels are particularly noisy.

    To assess the performance of the products generated by the SWOT-HR simulator, synthetic heights were compared to the
simulated H2D2 elevations, considered as the actual observations. For this purpose and, similar to Domeneghetti et al.



(2018), the SWOT-derived height pixels clouds were averaged over a window of 25 m by 25 m to estimate the error of the water levels. The mean error, as well as the root mean square error, were also calculated and used as global quality indicators of the heights derived from the SWOT raw data (i.e. pixel cloud).

## 4.3 Volume analysis

In order to remain true to the goal of the study to evaluate expected SWOT performance over lakes and not rivers, which have their own set of challenges, a mask was created to exclude the channels going in and out of the lake. Henceforth, WSE and water volumes refer to the area within the mask (Fig. 2) unless stated otherwise. The water balance of Mamawi Lake was performed for two types of data: SWOT imagery and in situ gauges

   The first type of data is from simulated SWOT imagery. WSE over the entire lake was retrieved in two ways. The first is

computed using the mean elevation of SWOT pixels, assuming the WSE to be uniform over the lake. The second is computed by passing a linear plane through available SWOT pixel cloud data, such that the square of the normal distance between the plane and the pixel cloud is minimized. This was performed for different hypothetical passes covering various areas of the lake in order to asses the impact of the spatial coverage on the water volume retrieval. Though the lake size (approximately 19 km at its widest orbital-wise) is smaller than the swath width (50 km), fractional coverage of the lake

could be obtained if the lake is located close to the nadir or far range. These potential coverage scenarios are represented in Fig. 4, where each color represents a coverage band in increments of 10 %.

   WSE difference or water volume between two hypothetical passes could be extracted and compared with the true values extracted from the H2D2 model. Since every wind scenario lasted 24 hours, the result of which is used as input in the simulator, it was assumed that the time between SWOT passes also corresponded to 24 hours. When a fractional coverage of

the lake was specified, for example 20 %, it was assumed that only 20 % of the lake was covered for the first pass, at the beginning of the simulation when wind speed is 0 m/s, and 20 % for the second pass after 24 hours of constant wind. Every combination of 20 % coverage cases were considered when retrieving water volumes. These are broken down in the following way: east and west coverage for time = 0h and time = 24 h, adding up to a total of 4 cases per coverage band. Different water volumes can be retrieved from these 4 samples and converted into average net flow by dividing the volume

by the time interval between the two hypothetical passes.

   The water volumes were also compared with volumes estimated using synthetic pressure gauges spread over the lake for benchmarking purposes. It was assumed that the gauges would normally be located in easily accessible sites. Therefore, 3 gauges were added near the main channels around the lake (Fig. 2).

   WSE from gauges were computed in two different ways. The simplest way consists of applying the synthetic gauge reading,

or the average readings for multiple gauges, over the entire lake, assuming a uniform elevation throughout. The second approach uses linear planes computed differently depending on the number of available gauges. For one available gauge, it was assumed that the lake had a uniform elevation of equal value to the one extracted from gauge #1 (see Fig. 2). This synthetic gauge is closest to an existing hydrometric network gauge of the Water Survey of Canada in the channel





connecting the west side of Mamawi Lake with Lake Claire. If two gauges were assumed to be available, a second gauge

was added by the largest branch of Mamawi Channel to the east. A linear plane was created using a vector connecting these two elevations, and a second vector perpendicular to the former vector. This plane was used to extrapolate over every point in the lake. Finally, when three gauges were assumed to be available, the gauge near the delta formed by Mamawi Creek to the south was also considered. A linear plane could be traced through each point and extrapolated throughout the rest of the lake. As with SWOT data, the water volume between passes was compared with the true values extracted from the H2D2

model.

Each gauge was assumed to have an unbiased normally distributed error of 0.5 cm centered around the true WSE value extracted from the hydrodynamic model. To take the gauge error into consideration, 100 samples were taken for each time step, creating as many pairs of linear planes for WSE.

## 5 Results and Discussion

### 5.1 Hydrodynamic simulations of the Mamawi Lake

A total of 25 simulations were performed using the H2D2 model. The first one is the scenario without wind (0 m/s), which is the starting point for all the other simulations. The remaining 24 wind scenarios include all 8 directions (cardinal and intercardinal points) applied to all the other wind speeds (5, 10 and 15 m/s). Examples are shown in Fig. 5.

The constant wind over 24 hours was sufficient to cause the water levels to adjust in a gradient following the direction of the

outgoing wind. The gradient becomes stronger as wind speed increases. Some exceptions can be noticed near the east and west channels, resulting from the water flowing over the banks from or into the nearby channels, which are omitted by the lake mask, by a more direct way than the main route. These areas are also strongly vegetated, with relatively high Manning coefficients, allowing for steeper slopes to form than on the open water lake area. Though the lake mask aimed to reduce this effect, it was not adjusted to exclude those regions. It is unknown whether or not this reflects the real lake behaviour of the

lake, but it is nonetheless the true state as simulated by H2D2. It also allows for some local heterogeneities, which can be found in real lakes from nearby channels for example, without being dominated by this effect.

### 5.2 SWOT height simulations

Twenty-five synthetic WSE maps for Mamawi Lake were generated using the SWOT-HR. An example of the classification for two different scenarios corresponding to a wind of 0 m/s and 15 m/s blowing southwards are presented in figs. 6a and 6b,

respectively. As observed, the classification of pixels is similar in both scenarios except for some areas (highlighted in red) where more significant differences are noticed. This is because the simulations by the SWOT-HR simulator are performed under a constant hypothetic average wind speed. For all the simulated scenarios, most pixels belong to the interior water class and a small number to the water near land edge class. The SWOT derived WSE are centered around a mean value of $209.77 \pm 0.37$ m for all scenarios when considering only wet pixels (classes 3 and 4). It must be taken into account that when





including the dark water class, the signal corresponds mainly to noise and therefore, the quality of the retrieved geolocated heights for these pixels can be notably deteriorated.

Some examples of the retrieved SWOT pixel cloud heights are shown in Fig. 7. They correspond to the same scenarios as in fig 5. Despite the water extent being similar across the scenarios, a notable spatial variation is observed for retrieved WSE. Note also that only the pixels corresponding to the water near land edge and the interior water classes are depicted. Larger

empty areas inside the lake correspond principally to dark water. Detection of these areas at low incident angles are challenging due to the low land-water contrast (Solander et al., 2016). Some of these pixels might correspond to dense areas of emergent vegetation or where the presence of algal beds is important (see Fig. 6), yet the procedures in the simulations are statistical and therefore mainly associated to dark water and do not reflect the actual physical reality. Similarly, some pixels inside the lake may be incorrectly classified as dry pixels (classes 1 and 2). Nevertheless, most of those pixels are generally

located at the shores of the lake and therefore expected to correspond to dry areas. From a visual inspection the distribution of the SWOT derived WSE are similar to those simulated by the H2D2 model, and particularly more visible for the speeds 10 m/s and 15 m/s. Overall, the largest over and underestimation of WSE occurred in scenarios corresponding to speeds > 5 m/s and to winds blowing southward and northward directions.

Fig. 8 shows the mean errors (ME) and the root mean square error (RMSE) for the derived water heights for the 25 simulated

scenarios. Figs. 8a and 8b depict the ME and RMSE for the surface elevations only corresponding to wet areas (classes: interior water and water near land), respectively. Dark water pixels might be used for the estimation of water extent but not for analysis that involve the simulated heights of these pixels, because, as previously stated, they are very noisy. The ME and RMSE values vary between ~0.05 and ~-0.02 meters and between ~0.3 and 0.7 meters, respectively. If the dry and the dark water areas are considered, the ME (> 0.6 meters) and the RMSE (> 2 meters) notably increased (figs. 8c and 8d). This result

is not surprising since the Ka-Rin band height retrieval algorithm is not expected to perform well on areas corresponding to land and/or darker surfaces (Solander et al., 2016; Domeneghetti et al., 2018). Some pixels affected by layover and other geometrical effects may also introduce important outliers, particularly those located near land, although they could be within the lake (interior water class).

The largest outliers in Fig. 8a correspond to the scenario where a wind coming from the northwest at a speed of 10 m/s is

simulated. The larger errors (ME: 0.05 m, RMSE: 0.7 m) for this scenario might be more important because of the orientation of the wind and the water movement with respect to the satellite track which is descending but principally from a misclassification of pixels. The direction of the water movement may create a slope (adding to the natural slope of the lake; Fig. 5a) which disposition with respect of the satellite viewing geometry might have the same effect of an embankment of a river in the near range. This disposition can produce layover and other geometrical distortions that may be more significant

when considering all the pixels classes. Points affected by these types of distortions are possibly affected by larger geolocation errors (e.g., Domeneghetti et al., 2018). Moreover, the source of large outliers can also come from the averaging of pixels and from the interpolation to resample the pixel-cloud into the grid of the elevations derived from the hydrodynamic model. Indeed, by inspecting the error maps, some of the larger errors are found at locations where the





interpolation is applied to fill gaps. Therefore, these errors might not correspond to the actual errors associated to the SWOT
products and direct comparison with the mission requirements may be not be reasonable, as suggested by Domeneghetti et
al. (2018). Other important mean errors are also identified in other incoming wind directions in figs. 8a ans 8b. For example,
a high RSME is noted (> 0.4 m) for the case of a wind blowing at 15 m/s and coming from the southwest direction.

### 5.3 Volume and net flow analysis

The difference between net flows retrieved from the synthetic gauges and true net flows are shown in Fig. 9a and Fig. 10a
for the uniform and linear plane approaches to extrapolate WSE, respectively. The results are aggregated according to the
number of available stations. Each box summarizes 100 samples to take into consideration the gauge error. As a reference,
the true average net flow for 5, 10 and 15 m/s wind scenarios are -0.09, -4.45 and -10.88 $m^3$/s, respectively, corresponding to
a net flow out of the lake. The inflow averages between 130 and 154 $m^3$/s, to put numbers into perspective.

As expected, net flow difference decreases as the number of gauges, or the amount of information available, increases. This
is the case for all wind speeds. Conversely, net flow difference increases as wind speed increases. Since the gauges are
stationed relatively close to the center of the lake, they are less susceptible to the change in water level caused by the wind.
The effect of local heterogeneities shown in Fig. 5 also have little effect on the results since they occur away from the
gauges. It should also be noted that when multiple gauges are available, using a linear plane to extrapolate WSE yields more
accurate retrievals at higher wind speeds, which is when the lake WSE becomes less uniform.

The patterns observed with synthetic gauges are also present in the flow difference analysis for SWOT retrievals (Fig. 9b and
Fig. 10b for uniform and linear plane approaches to extrapolate WSE, respectively). The results are aggregated according to
the percentage of lake area covered by SWOT passes. Each box summarizes the combinations of wind directions and lake
side covered: 2 sides (East or West end) for the first pass and 8 wind directions for each of the 2 sides of the lake that is
covered for the second pass. This makes a total of 2x2x8 = 32 scenarios for each % band of lake coverage.

In general, as wind speed increases, so does the error in net flow retrievals. As lake coverage increases, more information is
added, and the flow error decreases. This is the case for all wind speeds but amplified compared with gauges. This is partly
the result of the noise parameters specified in the SWOT-HR simulator, but also amplified by the local heterogeneities seen
in Fig. 5. Since these heterogeneities occur at the east and west ends of the lake, which is covered in the first or last 10 %
band, they have a large impact on water level interpolation, particularly for low lake coverage (see Fig. 4 for SWOT pass
coverage) and particularly if a linear plane is used to extrapolate WSE. For example, WSE retrieved from a first pass
covering only the first 10 % of the east end of the lake might be lower than the rest of the lake. As a result, WSE generated
assuming a uniform level would be underestimated. On the other hand, the linear plane might increase drastically toward the
west end due to the local slope, resulting in an increasingly underestimated retrieval as the local slope increases. If the
following pass after 24h only covers the first 10 % of the opposite (west) end, the WSE estimated from the uniform approach
would be overestimated, while the linear plane would decrease drastically toward the east end, resulting in an
underestimation of WSE. The resulting net flow would be positive with a uniform water level approach and negative using a



linear plane approach. Over- and underestimations are both present in the figures. As more lake area gets covered, the effect of these heterogeneities decreases. Similarly, this effect reduces as wind speed increases as the changes in lake WSE becomes more important.

However, unlike synthetic gauges, there is a flow retrieval bias for low area coverage, particularly for the linear plane approach. This is mainly caused by the asymmetric effect of wind on lake water levels. This is demonstrated in Fig. 11 for the retrieval of lake water volume at wind speeds of 15 m/s and 10 % lake coverage, using the linear plane approach as an example. The error for a given pass is not entirely symmetrical around the cardinal and intermediate directions. The largest differences in retrieved water volume generally occur when the incoming wind is in East-West axis, which is the same as the

general flow in the lake, and the partial coverage is in the opposite direction of the incoming wind. The average error for all passes combined is lowest when the wind is coming from the North, followed by the South, both of which are perpendicular to the natural flow. The effect of the wind on retrievals is not entirely symmetrical between passes, resulting in a bias seen in Fig. 10b.

When the SWOT coverage is low, retrievals from even a single synthetic point measurement are more reliable. However, as

SWOT coverage increases, the retrievals become increasingly precise, to the point where a 100 % lake coverage yields more reliable results than the 3 synthetic gauges combined.

## 6 Conclusions

The study, one of several ongoing SWOT-C TH (Pietroniro et al. 2019) compared water heights computed by the SWOT-HR hydrology simulator with those provided by the H2D2 hydrodynamic model under various wind scenarios. Resulting

estimates of water storage were also compared with those provided by gauges measurements within a synthetic framework. The analysis of the results highlighted the importance of having a high percentage of lake coverage included in SWOT passes, particularly in the presence of local heterogeneities. The accuracy of retrievals was also shown to depend on the method used to extrapolate WSE. An approach assuming uniform water levels was shown to be less prone to large errors, particularly for low lake coverage, compared with an approach using a linear plane which seems to produce better results for

a lake with 100% coverage. This is one of the methods included in the LOCNES toolbox developed by the CNES, which can be used to produce lakes products, such as WSE, extent and water storage (e.g, Pottier and Cazals, 2019).

The results presented are conditional to a number of assumptions and limitations. To begin, the study rests on the synthetic framework generated by hydrodynamic model and a number of parameters to drive it. The physically based nature of the model is assumed to reflect the general behaviour of a real scenario under similar circumstances. This excludes the

possibility of underground channel formations, water seeping out of the modelled lake aside from specified boundaries. It also does not consider evapotranspiration, which should be negligible compared with channel inflow and outflow over a period of 24 hours. Small waves normally generated under the influence of wind were not explicitly simulated at the scale used by the model, and their presence may influence SWOT height retrievals.



A similar argument can be applied to the SWOT-HR simulator, which was designed to represent expected images from a real

instrument including noises from various sources. However, known potential issues might need to be further analysed, such as dark water surfaces which can mislead the classification (e.g., Solander et al., 2016). Errors related to the geolocation due to geometrical distortions, DEM errors and other factors such as flooded vegetation need to be considered in the accuracy assessment. One possible solution to mitigate these types of errors would be through a post-processing of the pixel cloud by using the LOCNES. Additionally, the Radarsat Constellation Mission (RMC) which provide a higher frequency of

observations could also help to improve the water balance estimation by providing external complementary information (e.g., over flooded vegetation). Nevertheless, the use of a single SWOT pass with identical orbital parameters for every simulation also limits the generalizations of the conclusions.

Another assumption pertains to the WSE results obtained, which are specific to Mamawi Lake, including its size, shape and location of boundary conditions. It is expected that similar results would be obtained from lakes sharing similar

characteristics. Applying the conclusions of this study to lakes and reservoirs of different shapes and sizes could be considered if changes in WSE within the lake remain in good approximation to a plane. This is particularly important if only a small portion of the lake is covered by the SWOT image.

Finally, simulations under constant wind speeds and directions were performed in this study. This restriction resulted in lake water levels that are relatively well approximated by a linear plane. If wind conditions were not constant in strength or

direction, there may be stronger local heterogeneities in lake water levels that could lead to greater errors for partial SWOT coverage, just as it could for pressure gauges.

These assumptions and limitations represent challenges that may be used to stimulate future studies. These could include the introduction of nonuniform wind conditions, the use of longer or shorter periods between SWOT passes, using multiple SWOT passes with varying spatial coverages and orbital parameters such as varying incidence angle, evaluating the effect of

flooded vegetation or aquatic vegetation on SWOT retrievals, using more complex interpolation and extrapolation methods to retrieve lake water levels, as well as validation over a wider range of lake sizes and geometries.

Continued scientific progress is encouraged as the SWOT mission has the potential to provide unprecedented ability to monitor spatially-distributed channel, lake, and wetland WSE, such as the Peace-Athabasca Delta complex where the Wood Buffalo National Park Action Plan was developed in response to a UNESCO Reactive Mission Report that assessed potential

threats to the delta ecosystem. A novel satellite-based surface water monitoring approach would enhance addressing environmental flows -hydrology components that traditionally relied on sparse ground-based monitoring.

**Acknowledgement**

We thank Environment and Climate Change Canada and Parks Canada for their collaboration in this study. In particular, we thank David Campbell, Ronnie Campbell, Tom Carter, Mark Russell, Sébastien Langlois, Jessica Lankshear and Nicolas

Simard for help collecting data during the 2017 field campaign. We are also thankful for the tips granted by Pascal Matte and




Sébastien Langlois to set up the H2D2 model and improve its efficiency. We thank the CNES for providing access to the SWOT-HR simulator and processing tools, all developed by the SWOT JPL project. This research was funded by the Canadian Space Agency.

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



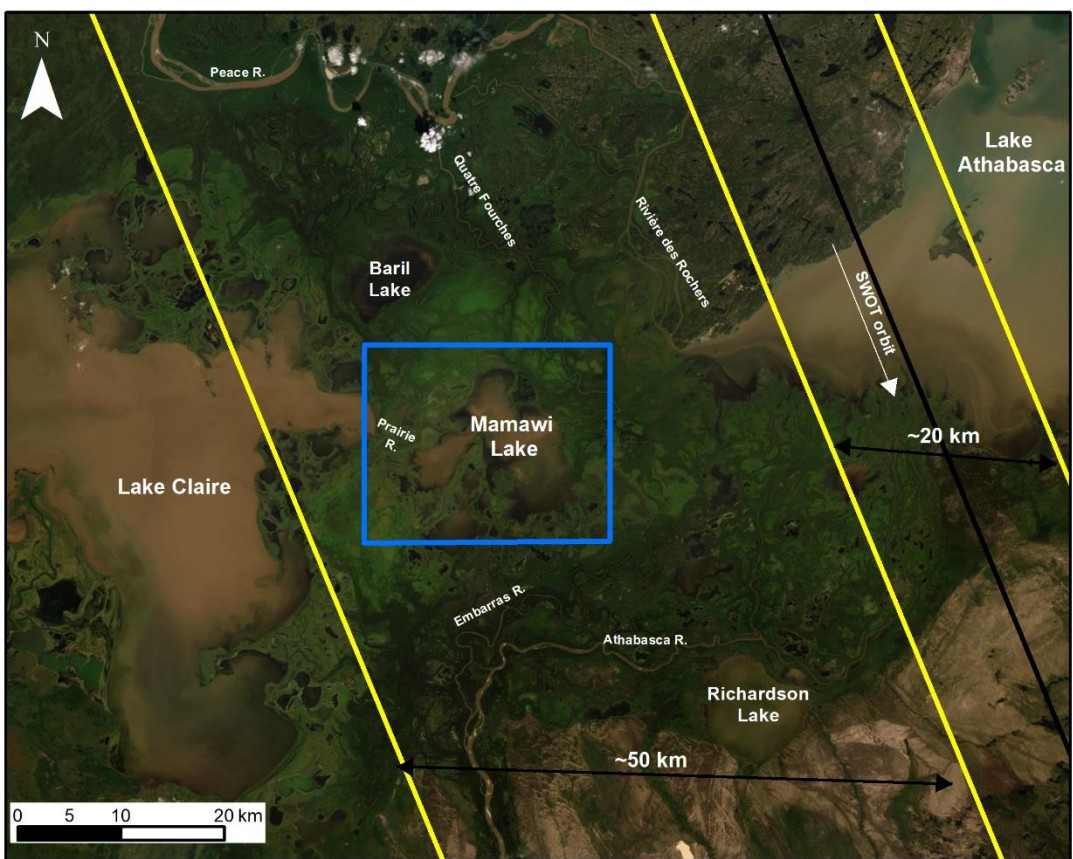

**Figure 1: Mamawi Lake (blue rectangle) and other main Lakes and rivers within the Peace-Athabasca Delta. The SWOT orbital**
**pass 303 over the area of study is depicted. The satellite nadir track is indicated by the black line and the satellite swath (~50 km)**
**delimited by the yellow lines. Basemap source: Esri, DigitalGlobe, GeoEye, Earthstar Geographics, CNES/Airbus DS, USDA,**
**USGS, AeroGRID, IGN, and the GIS User Community**





**Figure 2: Map of Mamawi Lake with the following elements : topobathymetry of Mamawi Lake - note that the channels are much deeper than the lake itself; Mamawi Lake water mask used to limit the extent of the hydrodynamic model (black line); inner lake contour used to computer water volumes (violet dashed line); water level observations used as boundary conditions in the hydrodynamic model (blue triangle); synthetic hydrometric station locations for point measurements (orange squares).**









**Figure 3: a) Backscattering image of Mamawi Lake taken from Sentinel-1 on July 8th, 2017; b) supervised classification of**
**Mamawi Lake using Sentinel-2 data taken on July 9th, 2017.**

**Figure 4: Hypothetical spatial coverage scenarios based on the expected SWOT pass #303 over Mamawi Lake.**





**Figure 5: A sample of the WSE results from the H2D2 simulations : a) no wind; b) 5 m/s wind coming from the East; c) 10 m/s wind coming from the South; d) 15 m/s wind coming from the North. Black areas represent no water. Note that the scale differs between figures for visibility purposes.**






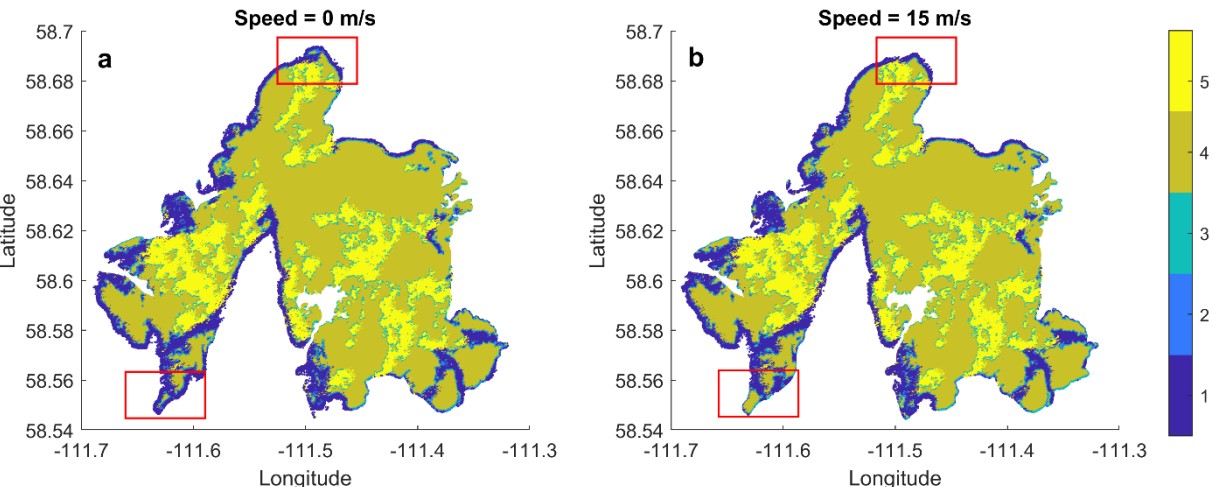

**Figure 6: Classification examples corresponding to two different scenarios. 1 : Land, 2 : Land near water, 3 : water near land edge, 4 : interior water, 5 : dark water**

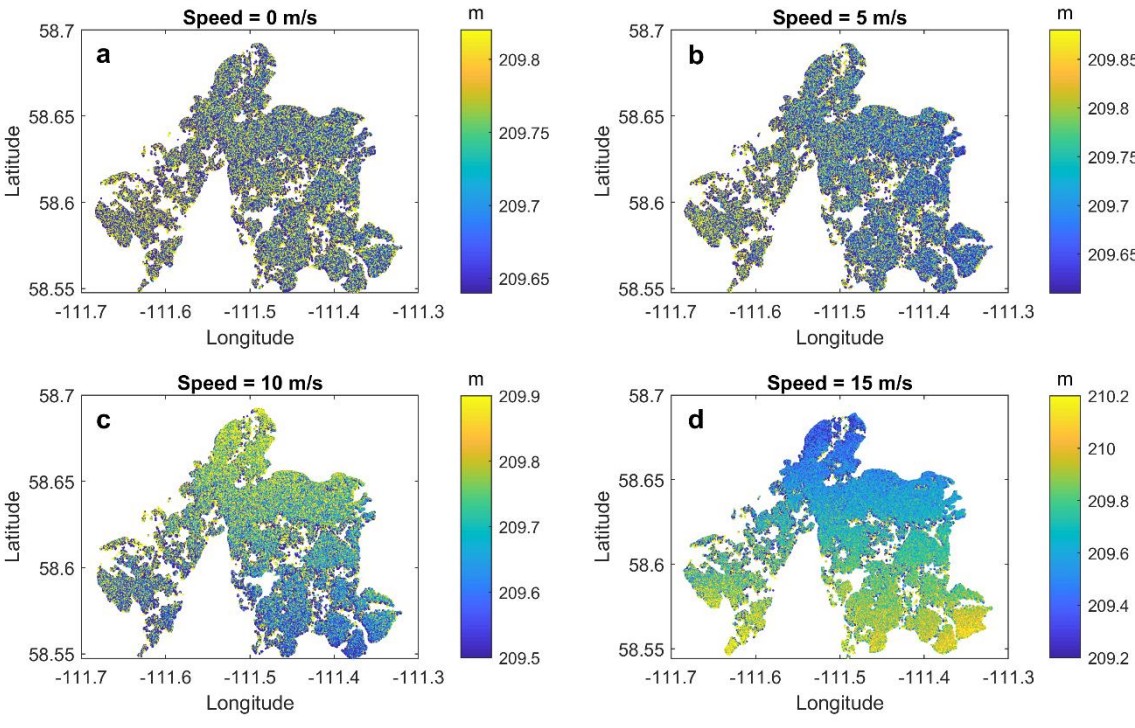


**Figure 7: A sample of the WSE results from the SWOT-HR simulator : a) no wind; b) 5 m/s wind coming from the East; c) 10 m/s wind coming from the South; d) 15 m/s wind coming from the North.**





**Figure 8: a) Mean and b) Root Mean Square errors on WSE as a function of incoming wind direction using wet pixels only. Similarly, c) and d) correspond to the mean error and RMSE including all pixel classes within the lake mask..**







**Figure 9: Net flow difference between flows retrieved from the true state and either a) synthetic gauge measurements, as a function of available station, or b) SWOT pixel cloud, as a function of lake area covered, assuming a uniform WSE. Every scenario begins with a windless case and ends after a 24h constant wind. All wind scenarios are merged into a single box.**





**Figure 10: Net flow difference between flows retrieved from the true state and either a) synthetic gauge measurements, as a function of available station, or b) SWOT pixel cloud, as a function of lake area covered, assuming the WSE follows a linear plane. Every scenario begins with a windless case and ends after a 24h constant wind. All wind scenarios are merged into a single box. It should be noted that the scales differ between a) and b), and that the b) plots are split using separate scales for lower (0-50 %) or higher (>50 %) coverage for visibility purposes.**





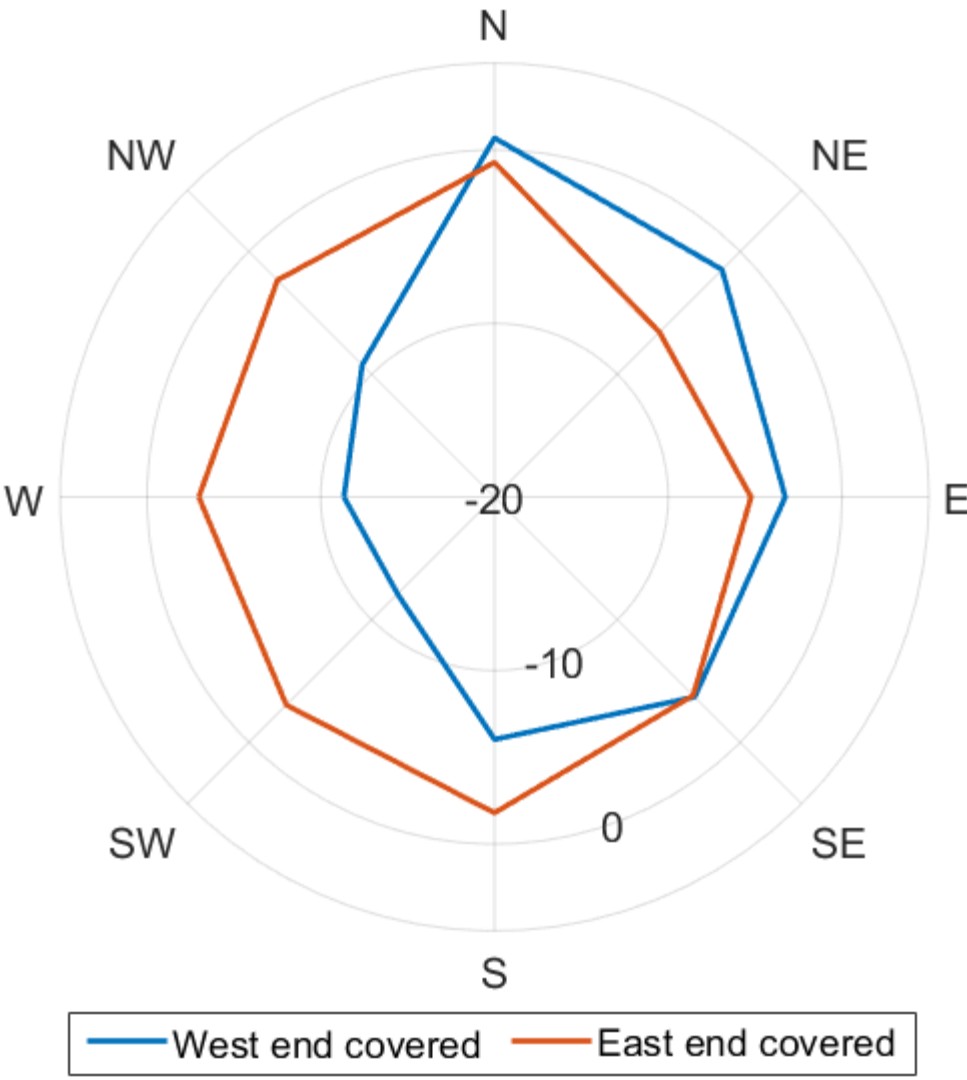

**Figure 11: Lake water volume difference between the true state and the volume retrieved from the SWOT pixel cloud using a linear plane to extrapolate WSE, as a function of incoming wind direction for a constant wind speed of 15 m/s over 24h. A lake coverage of 10 % is used. The blue curve represents a pass where only the west end of Mamawi Lake is covered, while the red curve represents a coverage of the east end.**