# Peer review of "Assessing the capabilities of the SWOT mission for large lake water surface elevation monitoring under different wind conditions"

_Hydrology and Earth System Sciences, 2020_

## Referee Comment (RC1) · Guy J.-P. Schumann (Referee) · 15 Jul 2020

This paper describes the capabilities of the soon-to-be-launched SWOT ocean/hydrology mission to measure lake water surface elevations under different wind conditions, using a high-resolution hydrodynamic lake model and the JPL SWOT hi-res simulator.

The paper is very well written and should be of interest to a wider audience. It is indeed a welcomed addition to the existing SWOT precursor studies that are mostly focusing on rivers with regard to the hydrology part of the mission.

[Figure]

I only have a few minor comments that should be addressed before publication.

- Paragraph L140: please describe briefly how the LiDAR DEM and SRTM DEM were merged/interpolated;

- L153: please specify which type of classification was used for the S-2 image;

- L160: Be sure to change the font type here to match the font type used throughout the manuscript;

- L202: Is the assumption of the noise being Gaussian justifiable? Please add some words about that here;

- L229: 4.3 section heading. Please specify what volume here. I suspect lake water volume?

- L270: section 5.1. Have you validated the model output in any way? I understand that for this synthetic case study, that may not be needed but it is nonetheless a good idea to indicate whether or not, model validation with independent data was performed and if so, how well did the model do; and if not, why not;

- L328: 5.3 section heading. Again, here, please specify that this is water volume;

- The error descriptions are sound and the conclusions reflect the results and analysis well.

---

## Referee Comment (RC2) · Anonymous Referee #2 · 21 Jul 2020

This article investigates the capabilities of SWOT to retrieve water surface elevation (WSE) over lakes under various wind conditions and SWOT spatial coverages. The effect of wind on lake WSE can be very important in some cases (several times the expected measurement error), and since lake WSE will be a main product provided by SWOT, it is essential to quantify these effects on the SWOT-derived SWE. The study is quite short, but very well presented and the results and analyses clearly lead to the conclusions drawn by the authors. I only have a few minor suggestions that the authors should be able to address before publication.

L261. Is there any other assumption concerning the second vector? Is it assumed

horizontal?

L286. Could you explain how the wind speed is accounted for in the SWOT-HR simulator? Is this to simulate ripples at the lake surface that may impact the SWOT signal? Also, it is not clear to me how a "constant hypothetic average wind speed" may cause the differences shown in Fig. 6.

Fig 5 and Fig 7. An arrow showing the direction of the wind for each simulation could be added in each subplot.

Fig 9 and Fig 10. Could you add a zero-line in each subplot?

---

## Author Response (AR1)

Included in this document are the authors' response to reviewer comments, as well as a marked up version of the manuscript.

Aside from minor editing of the text and some figures, the most important changes are twofold. First, the following sections were added to confirm to submission requirements:

- Code and data availability

- Author contribution

- Competing interest

The other important change includes clarifications of how the SWOT-HR simulator makes use of a fixed background wind field and a discussion of this impact on the results, particularly in terms of coherence between a synthetic experiment and the real world.

*Italicized text*: Reviewer's comment

AR: Authors' response

**Comments to Referee #1 (Guy J.-P. Schumann)**

*This paper describes the capabilities of the soon-to-be-launched SWOT ocean/hydrology mission to measure lake water surface elevations under different wind conditions, using a high-resolution hydrodynamic lake model and the JPL SWOT hi-res simulator.*

*The paper is very well written and should be of interest to a wider audience. It is indeed a welcomed addition to the existing SWOT precursor studies that are mostly focusing on rivers with regard to the hydrology part of the mission.*

*I only have a few minor comments that should be addressed before publication.*

*- Paragraph L140: please describe briefly how the LiDAR DEM and SRTM DEM were merged/interpolated;*

The DEM was produced and provided by Environment and Climate Change Canada. The way the two DEMs were merged is as follows: the LiDAR DEM was given priority over the SRTM DEM due to its higher horizontal and vertical resolution. However, since the LiDAR DEM covers only parts of the area used, SRTM DEM data was used to patch missing data. The SRTM was downscaled to the LiDAR resolution using bicubic interpolation. This has been clarified in the revised version of the manuscript.

*- L153: please specify which type of classification was used for the S-2 image;*

A supervised classification was made using the maximum likelihood algorithm. This has been clarified in the revised version of the manuscript.

*- L160: Be sure to change the font type here to match the font type used throughout the manuscript;*

The font has been changed to match the rest of the manuscript.

*- L202: Is the assumption of the noise being Gaussian justifiable? Please add some words about that here;*

Water level gauges come in many forms, each with their advantages and disadvantages. The current study assumed the use of pressure transducers, which were used during the measurement campaigns. The specs ensure an accuracy of ±0.5 cm, with ±1 cm considered extremes. This is mostly the result of noise from the sensor itself, including built-in electronics, which should be well approximated by a Gaussian curve. However, it does not consider environmental factors which can create a bias. One such example is a transducer slowly sinking in the loose sediments at the bottom of the river or lake, giving the false impression of an increase in water levels. This implies that a ±0.5 cm accuracy is likely an optimistic estimate of the real gauge error. However, while purposely introducing biases in the synthetic measurements potentially adds realism to the study, it was deemed to be a less valuable

benchmark against which to compare SWOT-HR estimations than bias-free synthetic measurements.

*- L229: 4.3 section heading. Please specify what volume here. I suspect lake water volume?*

The section has been renamed "Lake water volume analysis" in the revised version of the manuscript.

*- L270: section 5.1. Have you validated the model output in any way? I understand that for this synthetic case study, that may not be needed but it is nonetheless a good idea to indicate whether or not, model validation with independent data was performed and if so, how well did the model do; and if not, why not;*

This is a great point and in fact leads to the next planned step in the project.

The short answer is that no robust method was used to validate the hydrodynamic model output. It was ensured that the model does converge with the imposed boundary conditions and that the water balance does close at each step of the calibration procedure (within 0.36 % for the final step, for example) and for every simulation. Otherwise, most field measurements were taken on the same day as a snapshot so that there are no continuous measurements to compare against, aside from the boundary conditions themselves. All other measurements taken were used to define the bathymetry and set up the model and are therefore not independent.

We agree that more realism in a synthetic study can yield more robust conclusions. This is why the next step in the project is similar in that it also aims to evaluate SWOT's ability to close the water balance of lakes under windy conditions, but adding dynamic wind conditions over longer periods. This will require additional measurements from other campaigns, as well as the ability to specify changing winds in the SWOT-HR simulator. It is also planned to extend the study site to include a wider variety of lakes. The current manuscript is a first step toward this goal.

*- L328: 5.3 section heading. Again, here, please specify that this is water volume;*

The section has been renamed "Lake water volume and net flow analysis" in the revised version of the manuscript.

*- The error descriptions are sound and the conclusions reflect the results and analysis well.*

*Italicized text*: Reviewer's comment

AR: Authors' response

**Comments to Anonymous Referee #2**

*This article investigates the capabilities of SWOT to retrieve water surface elevation (WSE) over lakes under various wind conditions and SWOT spatial coverages. The effect of wind on lake WSE can be very important in some cases (several times the expected measurement error), and since lake WSE will be a main product provided by SWOT, it is essential to quantify these effects on the SWOT-derived SWE. The study is quite short, but very well presented and the results and analyses clearly lead to the conclusions drawn by the authors. I only have a few minor suggestions that the authors should be able to address before publication.*

*L261. Is there any other assumption concerning the second vector? Is it assumed horizontal?*

It is assumed that the perpendicular lines are isolines such that all points that lie on the same line share the same value (see image below).

[Figure]

*L286. Could you explain how the wind speed is accounted for in the SWOT-HR simulator? Is this to simulate ripples at the lake surface that may impact the SWOT signal?*

Wind speed does affect the simulated backscattered signal (sigma 0) through what would be the overall effect of ripples or waves, or lack thereof. Regions with little or no wind will therefore generate areas of dark water. However, there is currently a mismatch between the wind used for the hydrodynamic model and the SWOT-HR simulator. The SWOT-HR simulator uses a fixed wind field over every region in the world and not the wind speed specified in the hydrodynamic model. The mean wind speed value extracted for the region is used to generate spatially-correlated random wind fields, which in turn affect the backscattered signal. This is why scenarios like the zero wind speed one contains nearly as many pixels with data as the other scenarios. In reality, most of the lake would likely be mostly covered by dark water pixels for the no wind scenario, except for areas with vegetation. This is an important point to consider in the evaluation of real expected error. Additional clarifications have been added to sections 4.2, 5.2 and the conclusion of the revised manuscript to address this issue.

*Also, it is not clear to me how a "constant hypothetic average wind speed" may cause the differences shown in Fig. 6.*

The comment about the constant hypothetic average wind speed was meant to explain why the classification of pixels were similar, not why there were differences. The differences are caused by the different WSE generated by the different speeds, which can have a small impact on the classification through the different resulting angle of incidence for example. The sentences have been modified added in the updated version of the manuscript for clarity.

*Fig 5 and Fig 7. An arrow showing the direction of the wind for each simulation could be added in each subplot.*

An arrow showing the direction of the incoming wind has been added in the updated version of the manuscript.

*Fig 9 and Fig 10. Could you add a zero-line in each subplot?*

A zero-line has been added in the updated version of the manuscript.

[revised manuscript text omitted]

**Legend**

- ┄┄ Mamawi lake inner contour
- ☐ Hydrodynamic model limit
- 🟧 Synthetic station location
- 🔺 Boundary conditions

**Topobathymetry (m a.s.l.)**
- High : 230
- Low : 203

615

**Figure 2: Map of Mamawi Lake with the following elements : topobathymetry of Mamawi Lake - note that the channels are much deeper than the lake itself; Mamawi Lake water mask used to limit the extent of the hydrodynamic model (black line); inner lake contour used to computer water volumes (violet dashed line); water level observations used as boundary conditions in the hydrodynamic model (blue triangle); synthetic hydrometric station locations for point measurements (orange squares).**

[Figure]

Figure 3: a) Backscattering image of Mamawi Lake taken from Sentinel-1 on July 8th, 2017; b) supervised classification of Mamawi Lake using Sentinel-2 data taken on July 9th, 2017.

[Figure]

Figure 4: Hypothetical spatial coverage scenarios based on the expected SWOT pass #303 over Mamawi Lake.

[Figure]

625

**Figure 5: A sample of the WSE results from the H2D2 simulations : a) no wind; b) 5 m/s wind coming from the East; c) 10 m/s wind coming from the South; d) 15 m/s wind coming from the North. Black areas represent no water. Note that the scale differs between figures for visibility purposes. The red arrows point in the direction of incoming wind.**

[Figure]

Figure 6: Classification examples corresponding to two different wind scenarios. The legend is as follows: 1 = Land, 2 = Land near water, 3 = water near land edge, 4 = interior water, 5 = dark water. The red arrow points in the direction of incoming wind.

[Figure]

Figure 7: A sample of the WSE results from the SWOT-HR simulator : a) no wind; b) 5 m/s wind coming from the East; c) 10 m/s wind coming from the South; d) 15 m/s wind coming from the North. The red arrows point in the direction of incoming wind.

[Figure]

**Figure 8: a) Mean and b) Root Mean Square errors on WSE as a function of incoming wind direction using wet pixels only. Similarly, c) and d) correspond to the mean error and RMSE including all pixel classes within the lake mask..**

[Figure]

Figure 9: Net flow difference between flows retrieved from the true state and either a) synthetic gauge measurements, as a function of available station, or b) SWOT pixel cloud, as a function of lake area covered, assuming a uniform WSE. Every scenario begins with a windless case and ends after a 24h constant wind. All wind scenarios are merged into a single box.

[Figure]

**Figure 10: Net flow difference between flows retrieved from the true state and either a) synthetic gauge measurements, as a function of available station, or b) SWOT pixel cloud, as a function of lake area covered, assuming the WSE follows a linear plane. Every scenario begins with a windless case and ends after a 24h constant wind. All wind scenarios are merged into a single box. It should be noted that the scales differ between a) and b), and that the b) plots are split using separate scales for lower (0-50 %) or higher (>50 %) coverage for visibility purposes.**

[Figure]

**Figure 11: Lake water volume difference between the true state and the volume retrieved from the SWOT pixel cloud using a linear plane to extrapolate WSE, as a function of incoming wind direction for a constant wind speed of 15 m/s over 24h. A lake coverage of 10 % is used. The blue curve represents a pass where only the west end of Mamawi Lake is covered, while the red curve represents a coverage of the east end.**